# Factors associated with tuberculosis treatment initiation among bacteriologically negative individuals evaluated for tuberculosis: An individual patient data meta-analysis

Sun Kim[1]*, Melike Hazal Can[1], Tefera B. Agizew[2], Andrew F. Auld[3], Maria Elvira Balcells[4], Stephanie Bjerrum[5,6], Keertan Dheda[7,8,9], Susan E. Dorman[10], Aliasgar Esmail[7,8], Katherine Fielding[11], Alberto L. Garcia-Basteiro[12,13,14], Colleen F. Hanrahan[15], Wakjira Kebede[16,17], Mikashmi Kohli[18], Anne F. Luetkemeyer[19], Carol Mita[20], Byron W. P. Reeve[21], Denise Rossato Silva[22], Sedona Sweeney[23], Grant Theron[21], Anete Trajman[24,25], Anna Vassall[26], Joshua L. Warren[27], Marcel Yotebieng[28], Ted Cohen[29], Nicolas A. Menzies[1,30]

1 Department of Global Health and Population, Harvard T.H. Chan School of Public Health, Boston, Massachusetts, United States of America, 2 US Centers for Disease Control and Prevention, Gaborone, Botswana, 3 US Centers for Disease Control and Prevention, Lusaka, Zambia, 4 Infectious Disease Department, School of Medicine, Pontificia Universidad Católica de Chile, Santiago, Chile, 5 Department of Clinical Research, University of Southern Denmark, Odense, Denmark, 6 Department of Infectious Diseases, Copenhagen University Hospital, Rigshospitalet, Copenhagen, Denmark, 7 Centre for Lung Infection and Immunity, Division of Pulmonology, Department of Medicine and UCT Lung Institute, Cape Town, South Africa, 8 South African MRC Centre for the Study of Antimicrobial Resistance, University of Cape Town, Cape Town, South Africa, 9 Faculty of Infectious and Tropical Diseases, Department of Infection Biology, London School of Hygiene and Tropical Medicine, London, United Kingdom, 10 Medical University of South Carolina, Charleston, South Carolina, United States of America, 11 TB Centre, Department of Infectious Disease Epidemiology, London School of Hygiene and Tropical Medicine, London, United Kingdom, 12 ISGlobal, Hospital Clínic – Universitat de Barcelona, Barcelona, Spain, 13 Centro de Investigação em Saúde de Manhiça (CISM), Maputo, Mozambique, 14 Centro de Investigación Biomédica en Red de Enfermedades Infecciosas (CIBERINFEC), Barcelona, Spain, 15 Epidemiology Department, Johns Hopkins Bloomberg School of Public Health, Baltimore, Maryland, United States of America, 16 School of Medical Laboratory Sciences, Jimma University, Jimma, Ethiopia, 17 Mycobacteriology Research Center of Jimma University, Jimma, Ethiopia, 18 FIND, Geneva, Switzerland, 19 University of California San Francisco, San Francisco, California, United States of America, 20 Countway Library of Medicine, Harvard University, Boston, Massachusetts, United States of America, 21 DSI-NRF Centre of Excellence for Biomedical Tuberculosis Research and SAMRC Centre for Tuberculosis Research, Division of Molecular Biology and Human Genetics, Faculty of Medicine and Health Sciences, Stellenbosch University, Tygerberg, South Africa, 22 Faculdade de Medicina, Universidade Federal do Rio Grande do Sul, Porto Alegre, Brazil, 23 Faculty of Public Health and Policy, London School of Hygiene & Tropical Medicine, London, United Kingdom, 24 Federal University of Rio de Janeiro, Rio de Janeiro, Brazil, 25 McGill University, Montreal, Québec, Canada, 26 Centre for Mathematical Modelling of Infectious Diseases, London School of Hygiene & Tropical Medicine, London, United Kingdom, 27 Department of Biostatistics, Yale School of Public Health, New Haven, Connecticut, United States of America, 28 Division of General Internal Medicine, Department of Medicine, Albert Einstein College of Medicine, New York City, New York, United States of America, 29 Department of Epidemiology of Microbial Diseases, Yale School of Public Health, New Haven, Connecticut, United States of America, 30 Center for Health Decision Science, Harvard T.H. Chan School of Public Health, Boston, Massachusetts, United States of America

* sunkim1@hsph.harvard.edu

**Data Availability Statement:** The data included in this study are subject to restrictions imposed by

the Institutional Review Board (IRB) and require data use agreements for access; therefore, they are not publicly available. For access to the dataset, please contact the IRB of the Harvard T.H. Chan School of Public Health at irb@hsph.harvard.edu and refer to the respective authors listed in Table K in S1 Supplement.

**Funding:** This work was supported by the National Institute of Allergy And Infectious Diseases of the National Institutes of Health (Award Number U01AI152084 to SD). Other authors have no funding to declare for this study. The funders had no role in study design, data collection and analysis, decision to publish, or preparation of the manuscript.

**Competing interests:** AFL has received research grant support to their institution unrelated to this work from Cepheid, Gilead, GSK, Merck and ViiV. She has received consulting fees from Vir. She has received pharmaceutical and laboratory donations as research support to her institution from Cepheid, Hologic and Mayne Pharma. NAM receives research funding from NIH, CDC, CSTE, and the European Commission, and is a consultant with the Global Fund to Fight AIDS, TB, and Malaria. Other authors have no competing interests to declare.

**Abbreviations:** aOR, adjusted odds ratio; ART, antiretroviral therapy; CI, credible interval; IPD, individual patient data; OR, odds ratio; PCR, polymerase chain reaction; RDT, rapid diagnostic test; SSM, sputum smear microscopy; TB, tuberculosis; WHO, World Health Organization.

# Abstract

## Background

Globally, over one-third of pulmonary tuberculosis (TB) disease diagnoses are made based on clinical criteria after a negative bacteriological test result. There is limited information on the factors that determine clinicians' decisions to initiate TB treatment when initial bacteriological test results are negative.

## Methods and findings

We performed a systematic review and individual patient data meta-analysis using studies conducted between January 2010 and December 2022 (PROSPERO: CRD42022287613). We included trials or cohort studies that enrolled individuals evaluated for TB in routine settings. In these studies, participants were evaluated based on clinical examination and routinely used diagnostics and were followed for $\geq$1 week after the initial test result. We used hierarchical Bayesian logistic regression to identify factors associated with treatment initiation following a negative result on an initial bacteriological test (e.g., sputum smear microscopy (SSM), Xpert MTB/RIF).

Multiple factors were positively associated with treatment initiation: male sex [adjusted odds ratio (aOR) 1.61 (1.31, 1.95)], history of prior TB [aOR 1.36 (1.06, 1.73)], reported cough [aOR 4.62 (3.42, 6.27)], reported night sweats [aOR 1.50 (1.21, 1.90)], and having HIV infection but not on ART [aOR 1.68 (1.23, 2.32)]. Treatment initiation was substantially less likely for individuals testing negative with Xpert [aOR 0.77 (0.62, 0.96)] compared to smear microscopy and declined in more recent years. We were not able assess why clinicians made treatment decisions, as these data were not available.

## Conclusions

Multiple factors influenced decisions to initiate TB treatment despite negative test results. Clinicians were substantially less likely to treat in the absence of a positive test result when using more sensitive, PCR-based diagnostics.

## Author summary

### Why was this study done?

- Tuberculosis (TB) remains one of the leading causes of infectious disease death worldwide.

- Despite advancements in TB diagnostics, many diagnoses are still based on clinical judgment rather than bacteriological evidence.

- Understanding why clinicians decide to initiate TB treatment despite negative bacteriological test results can improve diagnostic accuracy and treatment outcomes.

**What did the researchers do and find?**

- We conducted a systematic review and meta-analysis of individual patient data from studies conducted between January 2010 and December 2022, where individuals were evaluated for TB.

- Key factors associated with initiating TB treatment after a negative bacteriological test included male sex, history of prior TB, reported cough and night sweats, and having HIV infection but not on antiretroviral therapy. Clinicians were less likely to initiate treatment if the initial test was a PCR-based diagnostic like Xpert MTB-RIF (as compared to sputum smear microscopy (SSM)).

**What do these findings mean?**

- The study identifies several factors that influence clinicians' decisions to treat for TB despite negative bacteriological test results.

- These findings can help refine TB diagnostic and treatment protocols, improving patient outcomes and enhancing public health strategies.

- More evidence is needed on clinicians' decision-making processes, which we did not assess in this study.

## Introduction

Tuberculosis (TB) remains a leading cause of infectious disease death worldwide [1], and a key strategy for accelerating TB elimination is to improve capacity for rapid and accurate diagnosis in high-burden countries [2]. Traditional TB diagnostics have major limitations, with sputum smear microscopy (SSM) failing to identify a substantial fraction of TB cases, and sputum culture requiring up to 8 weeks to return results. However, since 2010 the World Health Organization (WHO) has endorsed several new PCR (polymerase chain reaction)-based diagnostics with the potential to improve TB case detection, including the Xpert MTB/RIF (Xpert), Xpert MTB/RIF Ultra (Xpert Ultra), Truenat MTB, Truenat MTB Plus, and Truenat MTB-RIF Dx assays [3]. These tests combine rapid turn-around time and high sensitivity, enabling timely and accurate TB diagnosis [3,4].

Despite the potential of these new diagnostics, several studies have found limited effects on TB diagnoses and mortality following their introduction [5–13]. Evidence from programmatic settings suggests that clinical diagnosis (diagnosis based on clinical criteria alone, made when a bacteriological test result is unavailable or is negative) may partially explain this finding [14–17]. In many countries, clinical diagnosis represents a substantial fraction of notified TB cases despite the widespread adoption of Xpert, and in 2022 clinical diagnoses represented 38% of total global notifications for pulmonary TB [1]. If some of the individuals testing false-negative on an initial bacteriological test are subsequently treated based on clinical criteria, this may reduce the incremental impact achieved by adopting a more sensitive diagnostic. However, the widespread use of clinical diagnosis may also increase the number of individuals incorrectly treated for TB and overlook cases of drug-resistant TB, as studies of the performance of clinical

diagnosis suggest that the specificity of clinical algorithms can be low [18–20]. For certain types of tuberculosis, such as extrapulmonary and pediatric TB, clinical diagnosis may be the primary diagnostic approach.

As higher sensitivity diagnostics become more commonly used, it is useful to understand current practices around clinical diagnosis, and the factors that affect clinical decision-making when diagnostic test results are negative. These clinical decisions will affect the overall sensitivity and specificity of TB diagnostic algorithms, as well as determining the incremental health impact of new diagnostics. In this study, we conducted a systematic review of studies reporting diagnostic decisions and treatment initiation following a negative test result received as part of routine TB diagnosis. Using these data, we conducted an individual patient data (IPD) meta-analysis to identify the factors that affect clinicians' decisions to treat for pulmonary TB despite a negative test result.

## Methods

The target population for this study was individuals evaluated for pulmonary TB disease in routine clinical settings, who had received a negative result on an initial diagnostic test (e.g., smear microscopy, Xpert MTB/RIF). We conducted a systematic review to identify data sets describing the individual characteristics as well as the outcome of TB diagnosis (i.e., whether or not TB treatment was initiated) for individuals in this target population. The protocol was registered with PROSPERO: CRD42022287613 [21] and approved by the Institutional Review Board of the Harvard School of Public Health (IRB21-1488).

### Search strategy and selection criteria

Studies were identified by searching Medline/PubMed (National Library of Medicine, NCBI) and Embase (Elsevier, embase.com). Controlled subject vocabulary terms (i.e., MeSH, Emtree) were included when available and appropriate. The search strategies were designed and carried out by a health sciences librarian (CM). The publication date was limited to 2010 to 2022 in order to restrict the analysis to the period over which new TB diagnostics were being introduced. The exact search strategies are provided in **Text A** in S1 Supplement. We also contacted subject matter experts to identify ongoing or recently completed studies not identified in the database search.

Studies eligible for the review included randomized controlled studies or cohort studies (a) that enrolled individuals evaluated for TB after presenting for care at routine healthcare settings; (b) where treatment decisions were based on diagnostic tests in routine use in that setting (i.e., additional tests conducted for research purposes were not used); and (c) where participants were followed for least 1 week following the initial diagnostic test to record whether or not treatment was initiated. We excluded systematic reviews and studies of nonhuman subjects, latent TB, hospitalized patients, multidrug-resistant TB, and active case finding. We also excluded study participants younger than 18 years of age.

Authors SK and MC independently reviewed the titles and abstracts of each identified article, assessing them for inclusion or exclusion using Covidence (Veritas Health Innovation, Melbourne, Australia, available at www.covidence.org). During the second screening stage, full-text articles were obtained for all articles considered relevant or possibly relevant ("yes" or "maybe") by both reviewers based on the initial title and abstract review. The authors then independently evaluated each full-text article to determine its eligibility. SK and NM contacted the investigators of studies meeting the inclusion criteria to obtain de-identified patient-level data. This study is reported as per the Preferred Reporting Items for Systematic Reviews and Meta-Analyses (PRISMA) guideline (see **Table A** in S1 Supplement).

## Variables of interest

For each study data set, we extracted data on individual-level variables describing the type of initial test received (e.g., Xpert, Ultra, Truenat, SSM), age (18 years or older), sex, presence of TB-related symptoms (cough, fever, night sweats, weight loss), results for any non-bacteriological tests performed (e.g., chest radiography), HIV status, morbidity score (e.g., Karnofsky score ranging from 0 (dead) to 100 indicating no evidence of disease) if available, TB diagnosis, whether TB treatment was initiated, date of treatment initiation, date of testing, date culture result was returned (if applicable), and duration of follow-up. We also extracted contextual variables including calendar year, country, and type of clinic at which the patient was evaluated (primary, secondary). We excluded individuals with inconclusive or missing results for the initial diagnostic test.

After data extraction, we created a master list of variables available from each study. Relevant variables that could influence diagnostic decision-making were selected based on TB diagnostic algorithms and guidelines consolidated by WHO [3,22]. Given that each study has different variables and units, we selected common variables across studies for meta-analysis and converted variable types for consistency across studies (e.g., conversion of continuous variables to categorical variables for symptom durations (unit in weeks)). We collated the harmonized IPD into a single data set.

Our primary outcome was whether or not an individual initiated TB treatment following a negative SSM, Xpert, or Xpert Ultra result (i.e., the standard of care for initial TB testing in each setting at the time of the study). While some studies undertook additional investigational tests that were not part of the routine care, clinicians were blinded to these results (recorded in each trial report). Although most studies collected samples for sputum culture, we restricted our analysis to the period before culture results became available.

For studies that recorded a variable indicating whether or not treatment was provided on clinical grounds, we used this variable as our outcome measure. For all other studies, we defined clinical diagnosis as instances where treatment was initiated following negative initial test results but before culture results became available.

## Data analysis

IPD meta-analysis was performed via logistic regression, specified for the binary outcome of whether or not an individual initiated treatment as defined above. To do so, we employed a hierarchical Bayesian model with country random effects (see **Text B** in S1 Supplement) to account for country-specific differences in diagnostic practices not reflected in other variables [23,24]. For the primary analysis, we fit univariable and multivariable regression models considering age (18 to 30 years, 31 to 40 years, >40 years), sex (female, male), history of prior TB (no, yes, unknown), reported cough (yes, no), reported night sweats (yes, no), reported fever (yes, no), HIV status (negative, positive (not on ART), positive (on ART), unknown), test type (SSM, Xpert, Xpert Ultra), and calendar year. These variables were included in the primary analysis based on their availability in the majority of data sets.

We conducted 2 secondary analyses using variables not available for a subset of data sets. First, using the data sets that provided information on symptom duration, we fit a modified version of the regression model for the primary analysis, in which the binary variables for cough, fever, and night sweats were replaced by versions of these variables that each stratified the observations into one of 3 levels (none, less than 2 weeks, 2 weeks and above). Second, for the data sets containing chest X-ray results, we reran the regression model for the primary analysis with this additional variable (normal, abnormal, unknown).

As a robustness check, we re-estimated the results of the main analysis with 2 alternative regression specifications. First, we adopted an alternative outcome definition, in which clinical diagnosis was defined as treatment initiation within 7 days of the initial diagnostic test. While potentially excluding some clinical diagnoses, this stricter definition may reduce the risk of bias due to variation in the definition of clinical diagnosis adopted by each study. Second, we re-estimated results using a regression model in which the country random effects were replaced by study random effects.

Additionally, we conducted a sensitivity analysis using the probit model, complemented by cross-validation. To address concerns about heterogeneity, we performed stratified analyses based on diagnostic tests and HIV status. We also reran the analysis with a model that combined Xpert and Xpert Ultra into a single category of rapid diagnostic tests (RDTs) for comparison against SSM. All statistical analyses were performed in R (v.4.2.3) using the "brms" package (v.2.19.0) [25–27].

## Results

Our database search identified 4,286 potentially eligible studies. After removal of duplicates, this resulted in 3,428 unique references for screening. After review of title and abstract of those references, full-text screening was performed on 161 studies, with 51 eligible studies identified (Fig 1). Following communication with investigators for each study, we obtained data from 18 eligible studies. Six of these studies were excluded after initial data cleaning due to missing key variables or considering a different target population. The final data set included observations collected between 2011 and 2020, covering 13 countries across 12 studies. Most of these countries are classified as high-burden for TB by the WHO. Table 1 reports demographic and clinical characteristics for the full analytic sample, and **Table C** and **Text C** in S1 Supplement provide details of each included study.

### Primary analysis

The main analysis included data for 15,121 adults evaluated for pulmonary TB for whom the initial TB test was negative. Of these individuals, 477 were initiated on TB treatment following clinical diagnosis. Table 2 summarizes the meta-analysis results as odds ratios (ORs) and adjusted odds ratios (aORs) produced by univariable and multivariable regression models, respectively, representing the odd ratio of TB treatment initiation among individuals with a given factor compared to the reference category.

Based on the multivariable analysis, we identified statistically significant increases in the odds of TB treatment initiation associated with male sex (aOR 1.61 compared to female sex, 95% credible interval (CI): 1.31, 1.95), having a history of prior TB (aOR 1.36 compared to individuals without prior TB, 95% CI: 1.06, 1.73), having reported cough (aOR 4.62 compared to no cough, 95% CI: 3.42, 6.27), having reported night sweats (aOR 1.50 compared to no night sweats, 95% CI: 1.21, 1.90), and having HIV infection but not on ART (aOR 1.68 compared to HIV–negative, 95% CI: 1.23, 2.32).

In terms of the tests used for initial TB diagnosis, we found lower odds of treatment initiation for individuals who had received a negative result on Xpert (aOR 0.77 compared to diagnosis via SSM, 95% CI: 0.62, 0.96) and who had received a negative result on Xpert Ultra (aOR 0.57 compared to diagnosis via SSM, 95% CI: 0.30, 1.07), although the results for Xpert Ultra were not statistically significant. We also estimated declining rates of treatment initiation over time, controlling for other factors (aOR 0.81 for each additional calendar year, 95% CI: 0.74, 0.90).

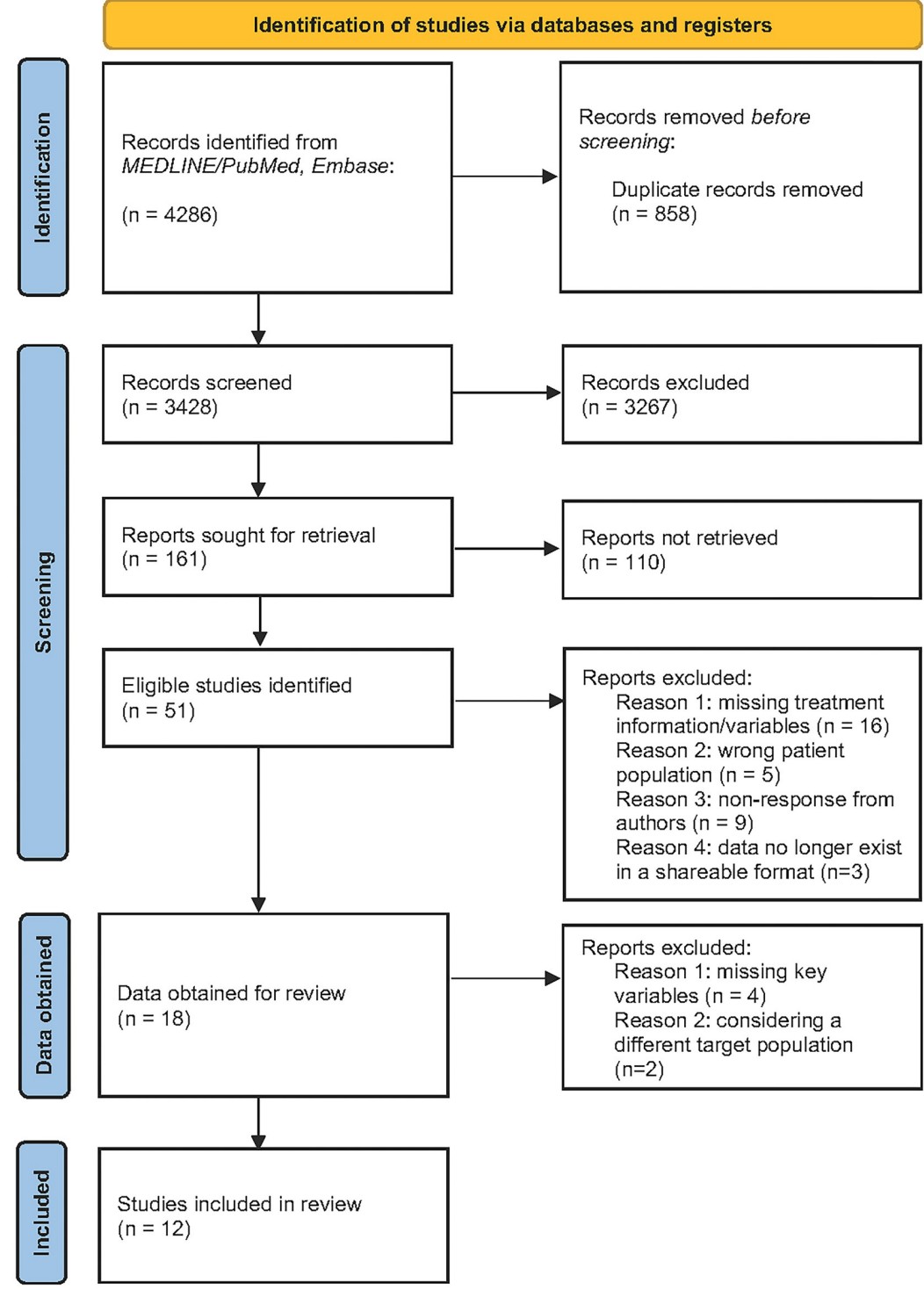

**Fig 1. Identification of studies and data to include in the meta-analysis.**

**Table 1. Demographic and clinical characteristics of study population.**

| | Included in study (*n*, column percentage) | Initiated on treatment (*n*, column percentage) | Percentage initiated on treatment (%) |
|---|---|---|---|
| Total sample | 15,121 | 477 | 3.15 |
| Age category | | | |
| 18–30 years | 4,884 (32.3%) | 116 (24.3%) | 2.38 |
| 31–40 years | 4,854 (32.1%) | 176 (36.9%) | 3.63 |
| 41 years and above | 5,383 (35.6%) | 185 (38.8%) | 3.44 |
| Sex | | | |
| Male | 6,081 (40.2%) | 263 (55.1%) | 4.32 |
| Female | 9,040 (59.8%) | 214 (44.9%) | 2.37 |
| History of prior TB | | | |
| Yes | 2,195 (14.5%) | 103 (21.6%) | 4.69 |
| No | 12,648 (83.7%) | 368 (77.1%) | 2.90 |
| Unknown | 278 (1.8%) | 6 (1.3%) | 2.16 |
| Reported cough | | | |
| None | 6,481 (42.9%) | 81 (17.0%) | 1.25 |
| Yes | 8,640 (57.1%) | 396 (83.0%) | 4.58 |
| Reported night sweats | | | |
| None | 10,341 (68.4%) | 222 (46.5%) | 2.15 |
| Yes | 4,780 (31.6%) | 255 (53.5%) | 5.33 |
| Reported fever | | | |
| None | 10,287 (68.0%) | 240 (50.3%) | 2.33 |
| Yes | 4,834 (32.0%) | 237 (49.7%) | 4.90 |
| HIV | | | |
| Negative | 3,101 (20.5%) | 90 (18.9%) | 2.90 |
| Positive, not on ART | 7,600 (50.3%) | 237 (49.7%) | 3.11 |
| Positive, on ART | 2,424 (16.0%) | 104 (21.8%) | 4.30 |
| Unknown | 1,996 (13.2%) | 46 (9.6%) | 2.30 |
| Study year | | | |
| 2011–2013 | 9,915 (65.6%) | 350 (73.4%) | 3.53 |
| 2014–2017 | 2,138 (14.1%) | 40 (8.4%) | 1.87 |
| 2018–2020 | 3,068 (20.3%) | 87 (18.3%) | 2.84 |
| Country | | | |
| Belarus | 97 (0.6%) | 1 (0.2%) | 1.03 |
| Botswana | 5,838 (38.6%) | 132 (27.7%) | 2.26 |
| Brazil | 272 (1.8%) | 5 (1.0%) | 1.84 |
| Ethiopia | 173 (1.1%) | 2 (0.4%) | 1.16 |
| Georgia | 300 (2.0%) | 6 (1.3%) | 2.00 |
| Ghana | 121 (0.8%) | 6 (1.3%) | 4.96 |
| India | 1,062 (7.0%) | 23 (4.8%) | 2.17 |
| Kenya | 290 (1.9%) | 16 (3.4%) | 5.52 |
| Papua New Guinea | 112 (0.7%) | 4 (0.8%) | 3.58 |
| Peru | 298 (2.0%) | 5 (1.0%) | 1.68 |
| South Africa | 5,877 (38.9%) | 168 (35.2%) | 2.86 |
| Uganda | 291 (1.9%) | 43 (9.0%) | 14.78 |
| Zimbabwe | 390 (2.6%) | 66 (13.8%) | 16.92 |

**Table 2. Odds ratios of TB treatment initiation following a negative diagnostic test result.**

| | Univariable analysis (95% Credible intervals) | Multivariable analysis* (95% Credible intervals) |
|---|---|---|
| Age category | | |
| 18–30 years | Ref | Ref |
| 31–40 years | 1.44 (1.13, 1.83) | 1.17 (0.92, 1.51) |
| 41 years and above | 1.42 (1.12, 1.80) | 1.11 (0.87, 1.43) |
| Sex | | |
| Female | Ref | Ref |
| Male | 1.80 (1.50, 2.16) | 1.61 (1.31, 1.95) |
| History of prior TB | | |
| None | Ref | Ref |
| Yes | 1.61 (1.13, 2.03) | 1.36 (1.06, 1.73) |
| Unknown | 0.81 (0.32, 1.77) | 0.73 (0.28, 1.65) |
| Reported cough | | |
| None | Ref | Ref |
| Yes | 5.93 (4.52, 7.88) | 4.62 (3.42, 6.27) |
| Reported night sweats | | |
| None | Ref | Ref |
| Yes | 2.34 (1.89, 2.89) | 1.50 (1.21, 1.90) |
| Reported fever | | |
| None | Ref | Ref |
| Yes | 1.84 (1.48, 2.28) | 1.13 (0.91, 1.39) |
| HIV | | |
| Negative | Ref | Ref |
| Positive, not on ART | 1.88 (1.38, 2.57) | 1.68 (1.23, 2.32) |
| Positive, on ART | 0.73 (0.52, 1.02) | 0.90 (0.64, 1.30) |
| Unknown | 0.98 (0.65, 1.46) | 0.82 (0.55, 1.20) |
| Diagnostic test | | |
| Sputum Smear | Ref | Ref |
| Xpert | 0.64 (0.51, 0.79) | 0.77 (0.62, 0.96) |
| Xpert Ultra | 0.21 (0.13, 0.34) | 0.57 (0.30, 1.07) |
| Year | 0.77 (0.71, 0.83) | 0.81 (0.74, 0.90) |

*Multivariable regression model also included country random effects, coefficients shown in **Table B** in S1 Supplement. Ref = reference category.

## Secondary analyses

In the first secondary analysis, we estimated ORs for cough, fever, and night sweats categorized by duration of symptoms, using data from the 5 studies for which this variable was available (7,468 observations). These findings indicated strong positive associations between TB treatment initiation and a reported cough of 0 to 2 weeks duration (aOR 3.29 compared to no reported cough, 95% CI: 1.64, 7.34) and >2 weeks duration (aOR 5.34 compared to no reported cough, 95% CI: 2.72, 11.82) (Fig 2). Reported night sweats of 0 to 2 weeks duration also demonstrated elevated odds of treatment initiation (aOR 1.45 compared to no reported night sweats, 95% CI: 1.06, 2.00).

The second secondary analysis estimated differences in treatment initiation based on chest X-ray result, using data from the 3 studies in which X-ray was conducted as part of TB

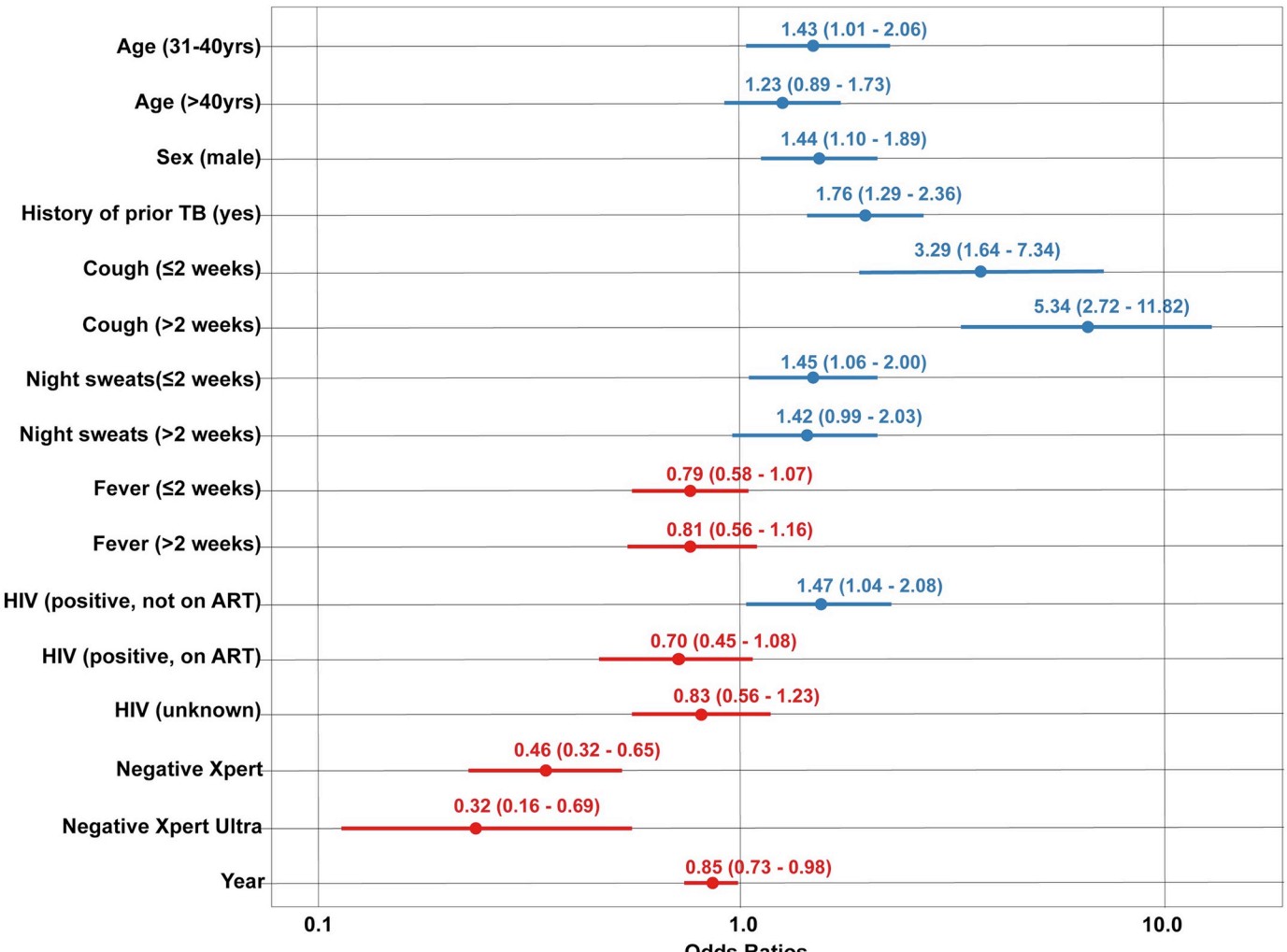

**Fig 2. Odds ratios of TB treatment initiation following negative diagnostic test result: Secondary analysis for data sets including duration of symptoms for cough, fever, and night sweats\* .\*** Reference group: Age 18–30 years old, female sex, no history of prior TB, no reported cough, no reported fever, no reported night sweats, HIV–negative, tested negative with SSM. Blue symbols signify ORs >1.0, red symbols signify ORs <1.0. ART, antiretroviral therapy; OR, odds ratio; SSM, sputum smear microscopy; TB, tuberculosis.

evaluation (2,449 observations). In these data, 1,677 individuals had a normal X-ray result (1.1% (18/1,677) initiated on treatment) and 456 had an abnormal X-ray result (6.1% (28/456) initiated on treatment). The results of this analysis showed that having an abnormal X-ray result had a strong positive association with treatment initiation, with an aOR of 6.89 (95% CI: 3.29, 14.42) compared to individuals with normal X-ray results (**Table D** in S1 Supplement).

## Alternative model specifications

Table 3 presents results for 2 alternative model specifications. In the first alternative specification, we analyzed an alternative outcome defined as treatment initiation within 7 days of the initial diagnostic test, representing 1.4% (205/15,121) of all observations. These results were generally consistent with the results of the primary analysis, although the odds ratio estimated for receiving a negative result on Xpert Ultra was lower than in the primary analysis and

**Table 3. Odds ratios of TB treatment initiation following negative diagnostic test result: Alternative model specification using strict outcome definition and study random effects.**

| | Adjusted odds ratios for treatment initiation (95% credible intervals) | | |
|---|---|---|---|
| | **Main analysis (primary outcome\*, country random effects)** | **First alternative specification (alternative outcome\*\*, country random effects)** | **Second alternative specification (primary outcome, study random effects)\*\*\*** |
| Age category | | | |
| 18–30 years | Ref | Ref | Ref |
| 31–40 years | 1.17 (0.92, 1.51) | 1.35 (0.90, 2.02) | 1.13 (0.88, 1.46) |
| 41 years and above | 1.11 (0.87, 1.43) | 1.37 (0.94, 2.06) | 1.07 (0.84, 1.37) |
| Sex | | | |
| Female | Ref | Ref | Ref |
| Male | 1.61 (1.31, 1.95) | 1.57 (1.17, 2.12) | 1.62 (1.34, 1.95) |
| History of prior TB | | | |
| None | Ref | Ref | Ref |
| Yes | 1.36 (1.06, 1.73) | 1.45 (1.03, 2.02) | 1.36 (1.08, 1.72) |
| Unknown | 0.73 (0.28, 1.65) | 0.57 (0.14, 1.75) | 0.39 (0.06, 1.65) |
| Reported cough | | | |
| None | Ref | Ref | Ref |
| Yes | 4.62 (3.42, 6.27) | 6.78 (3.78, 12.65) | 4.73 (3.50, 6.37) |
| Reported night sweats | | | |
| None | Ref | Ref | Ref |
| Yes | 1.50 (1.21, 1.90) | 1.29 (0.93, 1.78) | 1.45 (1.17, 1.82) |
| Reported fever | | | |
| None | Ref | Ref | Ref |
| Yes | 1.13 (0.91, 1.39) | 1.14 (0.84, 1.56) | 1.08 (0.87, 1.34) |
| HIV | | | |
| Negative | Ref | Ref | Ref |
| Positive, not on ART | 1.68 (1.23, 2.32) | 1.84 (1.16, 2.95) | 1.55 (1.12, 2.14) |
| Positive, on ART | 0.90 (0.64, 1.30) | 0.92 (0.58, 1.48) | 0.86 (0.59, 1.24) |
| Unknown | 0.82 (0.55, 1.20) | 0.38 (0.20, 0.69) | 0.87 (0.59, 1.29) |
| Diagnostic test | | | |
| Sputum Smear | Ref | Ref | Ref |
| Xpert | 0.77 (0.62, 0.96) | 0.65 (0.42, 0.98) | 0.79 (0.62, 0.99) |
| Xpert Ultra | 0.57 (0.30, 1.07) | 0.35 (0.17, 0.75) | 0.37 (0.21, 0.64) |
| Year | 0.81 (0.74, 0.90) | 0.97 (0.85, 1.09) | 0.87 (0.74, 1.04) |

\*Our primary outcome was the initiation of TB treatment after negative SSM, Xpert, or Xpert Ultra results; using treatment provision on clinical grounds if available, otherwise defining clinical diagnosis as treatment initiation post-negative initial tests prior to culture results.

\*\*For alternative outcome definition, clinical diagnosis was defined as treatment initiation within 7 days of the initial diagnostic test.

\*\*\*Multivariable regression model also included study random effects, coefficients shown in **Table E** in S1 Supplement. Ref = reference category.

statistically significant (aOR 0.35 compared to diagnosis via SSM, 95% CI: 0.17, 0.75). Additionally, the estimated time trend in treatment initiation was no longer significant (aOR 0.97 for each additional calendar year, 95% CI: 0.85, 1.09).

The results for the second alternative specification (results estimated with study random effects instead of country random effects) were generally consistent with the results of the

primary analysis, although the odds ratio estimated for receiving a negative result on Xpert Ultra was lower than in the primary analysis and statistically significant (aOR 0.37 compared to diagnosis via SSM, 95% CI: 0.21, 0.64). The estimated time trend in treatment initiation was no longer significant (aOR 0.87 for each additional calendar year, 95% CI: 0.74, 1.04).

Our sensitivity analysis results showed that our findings remain consistent across different models (logit versus probit) with cross-validation confirming that our main model exhibits a better fit (**Table F** in S1 Supplement). In addition, stratified analyses across different diagnostic tests and HIV status demonstrated the robustness and consistency of our results across various subgroups (**Tables G and H in** S1 Supplement). Further, the results combining Xpert and Xpert Ultra into a single category of RDT were also consistent with the findings of the primary analysis, as shown in **Table I** in S1 Supplement. Lastly, **Table J** in S1 Supplement provides additional evidence of the influence of each predictor on treatment decisions, reporting the absolute risks and risk differences of treatment initiation associated with a change in each predictor, holding others constant.

## Discussion

This study examined the factors associated with treatment initiation among adults evaluated for TB in routine healthcare settings, who had received a negative result on an initial bacteriological test for TB. Our analyses showed that male sex, a history of prior TB, reported cough, and having HIV infection but not receiving ART were positively associated with clinicians' decisions to initiate TB treatment. Among the 3 tests used for initial diagnosis, individuals receiving a negative result on Xpert were substantially less likely to be initiated on treatment compared to individuals who had received a negative result with SSM. Though not statistically significant in the main analysis, a negative result on Xpert Ultra was also associated with lower treatment initiation rates compared to SSM. In addition, the secondary analyses demonstrated increasing odds of treatment initiation with longer duration of cough (specifically, cough persisting for over 2 weeks). Similarly, the presence of an abnormal chest X-ray result was found to have a strong positive association with treatment initiation. We also observed a lower likelihood of treatment initiation in more recent years, controlling for other factors.

Most results from the alternative model specifications were consistent with those of the primary analysis. For the first alternative specification (outcome defined as treatment initiation within 7 days of the initial TB test), the fraction diagnosed clinically was lower than in the primary analysis (3.2% versus 1.4%), and this outcome definition may have excluded some individuals who were treated clinically but with a greater delay. However, this outcome definition reduced potential inter-study variation in the definition of clinical diagnosis, and the risk of bias due to clinicians accessing culture results before making treatment decisions. The second alternative specification assumed that residual variation in clinical decision-making was primarily attributable to study-specific factors (versus country-specific factors in the main analysis). That the estimated odds ratios were mostly consistent across different model specifications provides some assurance that these results are robust. One small difference was for Xpert Ultra, for which in both alternative specifications individuals testing negative on Xpert Ultra were estimated to be significantly less likely to begin treatment compared to those who received a negative result from SSM, with these odds ratios lower than those estimated in the primary analysis, and statistically significant. In addition, the results describing the time trend were no longer statistically significant in both alternative specifications.

The findings for individual covariates can be interpreted in light of factors that clinicians may consider during TB diagnosis. These considerations include the pre-test probability of disease (prevalence of TB disease among individuals being tested), the expected magnitude of

harms resulting from an incorrect negative diagnosis relative to the harms of an incorrect positive diagnosis, and the expected sensitivity and specificity of the tests being used. Several of the covariates examined in this study are relevant to these considerations.

First, several of the covariates we examined may influence clinician's beliefs about the pre-test probability of disease. Based on WHO guidelines for TB diagnosis and treatment in HIV-prevalent and resource constrained settings, a history of prior TB and symptoms suggestive of TB imply a higher pre-test probability of disease, and therefore may increase clinical suspicion for TB [28]. Similarly, in many settings persons living with HIV have higher TB incidence compared to HIV–negative individuals, and men have elevated incidence rates compared to women, such that clinicians may expect these characteristics to imply a higher disease prevalence among those evaluated for TB. In light of these relationships (each of which was linked to elevated treatment initiation rates), it is somewhat surprising that reported fever had a modest association with treatment initiation. While the presence of fever has been associated with TB, it is also associated with many other conditions, and therefore may be of limited value in distinguishing TB from other alternative diagnoses (as has been found with antibiotic trial as a diagnostic modality [29]).

For the second consideration (harms resulting from an incorrect negative diagnosis relative to the harms from an incorrect positive diagnosis), it is possible that this contributes to the elevated treatment initiation odds estimated for persons living with HIV, as compared to HIV–negative individuals. Individuals with both HIV and TB experience rapid disease progression and are less likely to survive the TB episode compared to HIV–negative individuals with TB [30–32]. As a consequence, the urgency of initiating TB treatment (if TB is suspected) will be much greater for individuals found to have HIV compared to those living without HIV. In contrast, the harms produced by a false-positive diagnosis, while not trivial, may not differ substantially between individuals with and without HIV.

For the third consideration (test sensitivity and specificity), this may explain the results estimated for the different test types (smear microscopy, Xpert, Xpert Ultra). The poor sensitivity of smear microscopy for pulmonary TB is well known, as is the improved performance of Xpert and Xpert Ultra compared to smear microscopy [33,34]. Because of the higher sensitivity of these new PCR-based tests, an individual testing negative on one of these tests is less likely to have TB than if the individual had instead tested negative with smear, all other things being equal. Clinicians aware of these relationships may be more hesitant to recommend treatment for patients that have tested negative with a high-sensitivity test. It is also true that each of the tests examined is known to have lower sensitivity among individuals with HIV infection [35], and this may be an additional factor contributing to the higher odds of treatment initiation for HIV–positive individuals following a negative test.

If it is true that clinicians are less likely to make a clinical diagnosis following a negative Xpert or Xpert Ultra result (versus a negative result on SSM), this could have implications for the impact of these new diagnostic tests on overall algorithm performance. Earlier modeling studies have demonstrated that clinical diagnosis could reduce the incremental effects of Xpert introduction on algorithm sensitivity, assuming that negative Xpert and negative SSM results are treated the same way by clinicians [36,37]. These effects could be magnified if clinicians are less likely to make clinical diagnoses following a negative Xpert, further reducing the overall impact of Xpert introduction on algorithm sensitivity, while at the same time increasing algorithm specificity. For individuals testing false-negative with Xpert, greater hesitance to initiate treatment based on clinical criteria could increase diagnostic delays, prolonging TB-related morbidity and mortality risks. Given the urgency of increasing TB case detection, further research on these potential mechanisms—and how to optimally balance the trade-offs involved in TB diagnosis—is needed.

There are several limitations to this study. First, we were not able to analyze all factors that potentially inform clinician decision-making, due to differences in the covariates recorded in the study data sets. It is possible that additional individual characteristics—such as recent weight loss or reporting a known TB contact, or medical comorbidities such as diabetes or other immunosuppressive conditions—may impact clinical decision-making but were not consistently captured in the study data. When X-ray results were available, they were found to have a major impact on clinical decision-making, but X-ray was only performed in a minority of studies. Similarly, it is possible that factors related to the healthcare setting, differences in national guidelines or protocols, or the capabilities of clinicians performing diagnosis may influence rates of clinical diagnosis. However, these setting-specific data were not available for analysis, contributing to differences in treatment initiation across countries. These setting-specific differences were substantial (quantified by the country random effects included in the main analysis), pointing to the existence of additional determinants of treatment initiation not captured by our analysis. Additional research to identify these factors is needed.

Second, our analytic population excluded patients aged under 18. While diagnosis for older children and adolescents may be similar to adults, clinicians will have different decision criteria for diagnosis of infants, due to both the different presentation of TB and the poor performance of available TB diagnostics in young children.

Third, while we selected studies to only include those performed under routine clinical conditions, it is possible that the behavior of clinicians performing TB diagnosis could have been influenced by their participation in clinical research. For example, it is possible that rates of clinical diagnosis will be lower in trial settings, if clinicians believe that missed diagnoses can be resolved through additional diagnostic testing undertaken as part of the trial (such as via sputum culture, performed in the majority of studies included in our review). If no additional testing is expected, clinicians may be more willing to make clinical diagnoses. Conversely, for routine settings where follow-up testing is common (or where multiple tests are conducted concurrently) rates of clinical diagnosis could be similar to those observed in trial settings. It is also possible that trial protocols may have influenced the factors considered during clinical diagnosis. Moreover, the clinics in which these studies were conducted may have been selected based on their capacity to participate in research, which may limit their representativeness of the general context of TB care.

Fourth, while many of the findings of the analysis are consistent with general principles of good patient care (as discussed above), we did not have access to additional evidence describing why clinicians made the decisions they did. Fifth, we did not compare clinical diagnosis decisions with culture results that subsequently became available. While such a comparison was outside the scope of the current study—which focused on clinical decisions made before any additional test results became available—this comparison would be useful for judging the diagnostic accuracy of clinical diagnosis and could be addressed in a subsequent study.

In conclusion, in this multi-country IPD meta-analysis of clinical diagnosis for TB, we found multiple clinical factors to be associated with the decision to initiate TB among individuals who receive a negative result on an initial bacteriological test for TB. Understanding these factors will allow for a more nuanced interpretation of the data describing the impact of introducing new TB diagnostics [37–39] and can inform efforts to refine clinical diagnostic algorithms, determine the appropriate balance between sensitivity and specificity when revising diagnostic approaches [40], and improve the overall performance of TB case detection.

## Supporting information

**S1 Supplement.** Table A. PRISMA Checklist. Table B. Demographic and clinical characteristics of participants, by study. Table C. Odds ratios of TB treatment initiation following negative diagnostic test result: country random effects from primary analysis. Table D. Odds ratios of TB treatment initiation following negative diagnostic test result: secondary analysis for data sets including chest X-ray results. Table E. Odds of TB treatment initiation following negative diagnostic test result: study random effects from alternative model specification. Table F. Sensitivity analysis comparing Logit vs. Probit model. Table G. Stratified analysis by diagnostic tests. Table H. Stratified analysis by HIV results. Table I. Odds ratios of TB treatment initiation following a negative diagnostic test result (Xpert/Xpert Ultra combined). Table J. Absolute risk and risk difference of treatment initiation. Table K. Contact information for accessing each data set included in the study. Text A. Search terms for Embase (Elsevier, embase.com). Text B. Hierarchical Bayesian logistic regression model. Text C. Description of individual studies included in analysis.
(PDF)

**S1 PROSPERO Protocol. PROSPERO: CRD42022287613.**
(PDF)

## Acknowledgments

The authors thank Cindy Imai for her support in the data retrieval process.

## Author Contributions

**Conceptualization:** Sun Kim, Ted Cohen, Nicolas A. Menzies.

**Data curation:** Sun Kim, Melike Hazal Can, Tefera B. Agizew, Andrew F. Auld, Maria Elvira Balcells, Stephanie Bjerrum, Keertan Dheda, Aliasgar Esmail, Katherine Fielding, Alberto L. Garcia-Basteiro, Colleen F. Hanrahan, Wakjira Kebede, Mikashmi Kohli, Anne F. Luetkemeyer, Carol Mita, Byron W. P. Reeve, Denise Rossato Silva, Sedona Sweeney, Grant Theron, Anete Trajman, Anna Vassall, Marcel Yotebieng.

**Formal analysis:** Sun Kim.

**Funding acquisition:** Susan E. Dorman, Ted Cohen, Nicolas A. Menzies.

**Investigation:** Sun Kim.

**Methodology:** Sun Kim, Joshua L. Warren, Nicolas A. Menzies.

**Project administration:** Sun Kim.

**Resources:** Sun Kim.

**Software:** Sun Kim.

**Supervision:** Ted Cohen, Nicolas A. Menzies.

**Validation:** Sun Kim, Melike Hazal Can.

**Visualization:** Sun Kim.

**Writing – original draft:** Sun Kim.

**Writing – review & editing:** Sun Kim, Melike Hazal Can, Tefera B. Agizew, Andrew F. Auld, Maria Elvira Balcells, Stephanie Bjerrum, Keertan Dheda, Susan E. Dorman, Aliasgar Esmail, Katherine Fielding, Alberto L. Garcia-Basteiro, Colleen F. Hanrahan, Wakjira

Kebede, Mikashmi Kohli, Anne F. Luetkemeyer, Carol Mita, Byron W. P. Reeve, Denise Rossato Silva, Sedona Sweeney, Grant Theron, Anete Trajman, Anna Vassall, Joshua L. Warren, Marcel Yotebieng, Ted Cohen, Nicolas A. Menzies.

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
