## [Editor Report · Decision Letter 0]

18 Apr 2024

Dear Dr Kim, 

Thank you for submitting your manuscript entitled "Factors associated with tuberculosis treatment initiation among bacteriologically negative individuals evaluated for tuberculosis: an individual patient data meta-analysis" for consideration by PLOS Medicine.

Your manuscript has now been evaluated by the PLOS Medicine editorial staff and I am writing to let you know that we would like to send your submission out for external peer review.

Please re-submit your manuscript within two working days, i.e. by the end of Monday 22nd April. If you need more time, please do just let me know.

Feel free to email the support team at plosmedicine@plos.org if you have any queries relating to your submission and uploading metadata. Otherwise, please feel free to contact me directly (ssunny@plos.org).

Kind regards,

Syba Sunny, MBBS, MRes, FRCPath

Associate Editor

PLOS Medicine

ssunny@plos.org

---

## [Decision Letter · Decision Letter 1]

3 Jun 2024

Dear Sun Kim,

Many thanks for submitting your manuscript "Factors associated with tuberculosis treatment initiation among bacteriologically negative individuals evaluated for tuberculosis: an individual patient data meta-analysis” (PMEDICINE-D-24-01238R1) to PLOS Medicine. The paper has been reviewed by three subject experts and a statistician; their comments are included below.

As you will see, the reviewers regarded your paper favourably. However, there were requests for more information and clarity in the manuscript, and one reviewer raised concerns about bias in the study and about the methodology used. After discussing the paper with the editorial team and an academic editor with relevant expertise, I’m pleased to invite you to revise the paper in response to the reviewers’ comments. We plan to send the revised paper to some of all of the original reviewers*, and of course we cannot provide any guarantees at this stage regarding publication.

When you upload your revision, please include a point-by-point response that addresses all of the reviewer and editorial points, indicating the changes made in the manuscript and either an excerpt of the revised text or the location (eg: page and line number) where each change can be found. Please submit a clean version of the paper as the main article file and a version with changes marked should as a marked-up manuscript. Please also check the guidelines for revised papers at http://journals.plos.org/plosmedicine/s/revising-your-manuscript for any that apply to your paper.

We ask that you submit your revision by June 24th. However, if this deadline is not feasible, please contact me by email, and we can discuss a suitable alternative.

Please don’t hesitate to contact me directly with any questions (ssunny@plos.org). If you reply directly to this message, please be sure to ‘Reply All’ so your message comes directly to my inbox.

Kind regards,

Syba

Syba Sunny MBBS, MRes, FRCPath

Associate Editor 

PLOS Medicine

ssunny@plos.org

*Please note: If your article is accepted, you may have the opportunity to make the peer review history publicly available. The record will include editor decision letters (with reviews) and your responses to reviewer comments. If eligible, we will contact you to opt in or out.

Editorial comments:

The editorial team read this manuscript with interest and thank the authors for choosing to submit it to PLOS Medicine. However, we agree with the reviewers’ comments and with those of the Academic Editor. Please note that these will need to be addressed in full to process to the next stage in the process towards publication.

1) Data Availability:

PLOS Medicine requires that the de-identified data underlying the specific results in a published article be made available, without restrictions on access, in a public repository or as Supporting Information at the time of article publication, provided it is legal and ethical to do so. Please see the policy at 

http://journals.plos.org/plosmedicine/s/data-availability

If the data are not freely available, please describe briefly the ethical, legal, or contractual restriction that prevents you from sharing it. Please also include an appropriate contact (web or email address) for inquiries (this cannot be a study author).

2) Reporting guidance:

Please report your study according to the relevant guidance which can be found here https://www.equator-network.org/reporting-guidelines/

3) Statistical reporting: (Please note that not all of this guidance will apply.)

Please quantify the main results with 95% CIs and p values.

When reporting p values please report as <0.001 and where higher as p=0.002, for example. When reporting 95% CIs please separate upper and lower bounds with commas instead of hyphens as the latter can be confused with reporting of negative values.

Please include the actual amounts and/or absolute risk(s) of relevant outcomes (including NNT or NNH where appropriate), not just relative risks or correlation coefficients. (example for absolute risks: PMID: 28399126).

Please include any important dependent variables that are adjusted for in the analyses.

4) Prespecified analysis plan/study protocol:

Did your study have a prospective protocol or analysis plan? Please state this (either way) early in the Methods section.

5) Abstract layout:

Please structure your abstract using the PLOS Medicine headings (Background, Methods and Findings, Conclusions).

6) Author summary:

At this stage, we ask that you include a short, non-technical Author Summary of your research to make findings accessible to a wide audience that includes both scientists and non-scientists. The authors summary should consist of 2-3 succinct bullet points under each of the following headings:

• Why Was This Study Done? Authors should reflect on what was known about the topic before the research was published and why the research was needed.

• What Did the Researchers Do and Find? Authors should briefly describe the study design that was used and the study’s major findings. Do include the headline numbers from the study, such as the sample size and key findings. 

• What Do These Findings Mean? Authors should reflect on the new knowledge generated by the research and the implications for practice, research, policy, or public health. Authors should also consider how the interpretation of the study’s findings may be affected by the study limitations. In the final bullet point of ‘What Do These Findings Mean?’, please describe the main limitations of the study in non-technical language.

The Author Summary should immediately follow the Abstract in your revised manuscript. This text is subject to editorial change and should be distinct from the scientific abstract. Please see our author guidelines for more information: https://journals.plos.org/plosmedicine/s/revising-your-manuscript#loc-author-summary

Comments from the reviewers:

Reviewer #1: 

The authors undertook an individual-participant meta-analysis for evaluating factors associated with treatment initiation among patients with negative tests for tuberculosis. The manuscript is well written and I only have a few comments.

1. The authors should provide a more detailed discussion on how rates of treatment initiation might differ between those observed within clinical trials, observational studies and routine clinical practice. Currently it is only briefly touched upon in the limitations section.

2. Discuss in more detail the differences in treatment initiation between various countries.

3. How did the authors account for multiple testing using the same test (e.g. studies that used only one sputum Xpert vs. studies that tested 3 sputum samples with Xpert). Discuss how this may influence treatment initiation among those negative.

Reviewer #2: 

Dear authors

With pleasure I read your manuscript. It reads very well. 

Please find some recommendations:

Abstract: 

Background: "after a negative diagnostic test result." please consider "after a negative bacteriological diagnostic test result. This will show you do not consider other tests, such as CXR.

"Understanding factors associated with clinicians' decisions to initiate treatment for individuals with negative test results is critical for predicting the potential impact of new diagnostics." Is not that self-explanatory to me. I would keep this for the main text. In the intro I would emphasize the knowledge gap, not the potential impact.

Main text

Intro:

"Compared to SSM and culture": no true comparison is made. Please reword.

"If some of the individuals testing false-negative on an initial bacteriological test are subsequently treated based on clinical criteria, this may reduce the incremental impact achieved by adopting a more sensitive diagnostic. However, the widespread use of clinical diagnosis may also increase the number of individuals incorrectly treated for TB and overlook cases of drug-resistant TB" Consider adding that the roll-out of a good -but still imperfect- test may delay diagnosis as clinicians may be hesitant to start TB treatment based on clinical criteria.

"be the sole option": I think this should be "the main option", as some patients (children, patients with EP TB) are bacteriologically diagnosed.

Methods: 

"The target population for this study was individuals evaluated for pulmonary TB disease in routine clinical settings, who had received a negative result on an initial diagnostic test (e.g., smear microscopy, Xpert MTB/RIF). We conducted a systematic review to identify datasets describing the individual characteristics as well as the outcome of TB diagnosis (i.e., whether or not TB treatment was initiated) for individuals in this target population." What if after an initial test a follow-up test was done? Then diagnosis is not based on clinical ground, neither on the initial test. 

The same comment applies to "(b) where treatment decisions were based on diagnostic tests in routine use in that setting (i.e., additional tests conducted for research purposes were not used)". In routine care patients are tested often repeatedly. Would such a patient or study be eligible, and which test result would be considered as the initial test? If you would exclude those with a repeat positive test, you could have some selection? If you allow studies where more than one test was, the meaning of having a neg bact test would not be the same, as patients would have a lower post-test prob of TB.

"While some studies undertook additional investigational tests clinicians were blinded to these results." Unclear to me. Your outcome "whether or not an individual initiated TB treatment" does not show whether a clinician was blinded for any follow-up result.

The categorization of age suggests that children were excluded. Was this shown above, in the methods? If so, I missed it.

Results: very clear

Discussion: very clear

Please add a paragraph on "Understanding factors associated with clinicians' decisions to initiate treatment for individuals with negative test results is critical for predicting the potential impact of new diagnostics" 

Reviewer #3: 

General comments:

This study presents a individual participant data meta-analysis exploring factors that associate with a decision to commence treatment for TB when rapid diagnostic tests (smear microscopy and NAATs) are negative.

The analysis includes data from 12 clinical cohort and randomised clinical studies, with a large sample size that is largely representative of patients encountered in high TB burden settings.

The study provides a useful overview of factors that associate with and are therefore presumed to influence the clinical suspicion of TB and decision to give treatment. In this respect it provides a review of physician behaviour and a measure of pre-test clinical probability of a diagnosis of TB that may be useful to understand the utility of existing and novel diagnostics. One example is presented in the paper of the effect of a negative Xpert or Xpert-ultra test on the clinician's assessment of TB likelihood. 

Specific comments:

1. As the data is collected from participants recruited to research studies, clinical decision making is likely to have been influenced by the operating study protocols. These protocols are likely to include many of he factors identified here as risk factors for TB and as such the risk of incorporation bias cannot be excluded. Whilst the dataset is large, it is therefore difficult to know how representative these findings are of routine clinical practice.

2. A significant limitation of the analysis, acknowledged by the authors, is that it is not underpinned by evidence to support or refute the appropriateness of the decision to start or withhold treatment for TB. The authors talk about the importance of avoiding both over-treatment and under-treatment in the absence of confirmatory diagnostics, but this manuscript does not provide us with information to draw any conclusions in this regard. Supporting evidence in the form of either bacterial confirmation where this data was available or a clinical and radiological response to treatment when culture was negative could provide useful insight of what may or may not be important. 

3. Furthermore, although mycobacterial culture is not routinely performed in high burden setting, it would be interesting and relevant to know whether the decision to delay starting treatment (>7days) differed between sites where culture was and was not performed. From a clinical standpoint, it could be entirely reasonable to wait for culture results in cases with an intermediate probability of TB. 

Reviewer #4: 

This paper is concerned with an interesting problem, the factors affecting the decision to treat TB following a negative test. 

Are there any unmeasured confounders reported in the literature that were not available in this analysis? 

Also, are there any other factors that clinicians tend to utilise in the diagnosis process that may be hard to quantify and therefore include in the analysis? 

Any other data completeness issues, especially regarding how representative are the present data of the overall picture? 

The authors have identified 51 studies as eligible, but only obtained the data from 18, is this an issue regarding the generalisation of the findings?

The student-t priors reported on supplement page 32 seem like a reasonable choice. However, it is well known that the PSRF criterion can be misleading since it is necessary for its value to be close to 1 but this is not a sufficient indication of algorithmic convergence. 

I suggest that the authors add the effective sample size, a more reliable measure of the quality of the algorithm (and its potential issues), summarising how many of those 4000 draws are roughly independent samples from the posterior density of interest.

The basic model seems like a reasonable choice. Did the authors try any alternatives as sensitivity analysis? Say using probit or cloglog links? Or perhaps ensemble-based like random forests or XGB?

This would also help to settle unclear findings like the effect of a negative result on Xpert Ultra.

How representative are these data overall? PloS Medicine takes a global view on public health and therefore this is an important discussion point.

Including a country or study specific random effect term sounds reasonable.

Are there any reasons to assume that these countries may not be "similar" and therefore exchangeable, perhaps initiating a discussion on accounting for some systematic difference?

Why exclude multi-drug resistant TB which represents an important problem? Please discuss.

[LINK]

Comments from the Academic Editor:

The Academic Editor was in agreement with the reviewers’ comments and recommended that these be addressed in full. He commented that, whilst the study was reasonably well done in many respects, that the insights offered were, at present, limited. He suggested that it would be useful if the analyses could be more robust in order to provide further depth to the results and to address concerns about heterogeneity. He suggested that the authors consider stratified analyses across diagnostic tests and perhaps HIV, too. He also suggested considering alternative ways to assess the value of the different TB tests, e.g. by calculating absolute risk of being treated with different patient profiles (e.g. HIV status, cough, night sweats) and different diagnostics. He also wondered whether there might be some potentially interesting results should you consider analyses regarding trade-offs, e.g. an Xpert Ultra result is equal to cough lasting more than 2 weeks + night sweats in determining whether healthcare workers initiate treatment or not.

1. Please upload any figures associated with your paper as individual TIF or EPS files with 300dpi resolution at resubmission; please read our figure guidelines for more information on our requirements: http://journals.plos.org/plosmedicine/s/figures. While revising your submission, please upload your figure files to the PACE digital diagnostic tool, https://pacev2.apexcovantage.com/. PACE helps ensure that figures meet PLOS requirements. To use PACE, you must first register as a user. Then, login and navigate to the UPLOAD tab, where you will find detailed instructions on how to use the tool. If you encounter any issues or have any questions when using PACE, please email us at PLOSMedicine@plos.org.

To submit your revised manuscript please use the following link:

---

## [Decision Letter · Decision Letter 2]

10 Oct 2024

Dear Dr. Kim,

Thank you very much for re-submitting your manuscript "Factors associated with tuberculosis treatment initiation among bacteriologically negative individuals evaluated for tuberculosis: an individual patient data meta-analysis" (PMEDICINE-D-24-01238R2) for review by PLOS Medicine.

I have discussed the paper with my colleagues and the academic editor and it was also seen again by 3 reviewers. I am pleased to say that, provided the remaining comments from the reviewers and the academic editor are addressed, and that editorial and production issues are dealt with, we are planning to accept the paper for publication in the journal.

[LINK]

Please also check the guidelines for revised papers at http://journals.plos.org/plosmedicine/s/revising-your-manuscript for any that apply to your paper. 

We expect to receive your revised manuscript within 2 weeks (which is an extended timeline, bearing in mind the number of requests that need addressing). Please email us (plosmedicine@plos.org) if you have any questions or concerns.

We look forward to receiving the revised manuscript by Oct 24 2024 11:59PM. 

Sincerely,

Syba Sunny, MBBS, MRes, FRCPath

Associate Editor 

PLOS Medicine

ssunny@plos.org

COMMENTS FROM THE ACADEMIC EDITOR:

The academic editor supported moving this manuscript forward to a minor revision outcome. However, he did make one request. He asked that the authors provide absolute risk and risk difference for at least the main analysis. In Table S8, the authors appear to state that they have done this, but it appears that only percentage change and risk difference were given. He kindly requests to see the absolute risk (i.e. 15% of PLWH not on ART with negative test received TB treatment vs. 10% of HIV negative individuals). He added that there should be no reference for absolute risk.

COMMENTS FROM THE REVIEWERS:

Reviewer #1: Thank you for providing a revised version of the manuscript. The authors have fully addressed all my comments. I have no further comments

Reviewer #2: Dear authors

Almost all comments were addressed. 

One of my comments might not have been clear. 

First I show the history of our exchange, then I re-formulate my comment, with a few more words.

In the manuscript: "If some of the individuals testing false-negative on an initial bacteriological test are subsequently treated based on clinical criteria, this may reduce the incremental impact achieved by adopting a more sensitive diagnostic. However, the widespread use of clinical diagnosis may also increase the number of individuals incorrectly treated for TB and overlook cases of drug-resistant TB" 

My comment was: Consider adding that the roll-out of a good -but still imperfect- test may delay diagnosis as clinicians may be hesitant to start TB treatment based on clinical criteria.

Your response was: We agree with the reviewer that this scenario could occur, but feel it is too hypothetical to include here as such a test was not considered in our study.

My follow-up comment: All TB diagnostic tests available at present are imperfect, especially with regards to sensitivity. Hence, my comment was not based on hypothetical grounds. Since the implementation of Xpert MTB/RIF, an imperfect test (with regards to its sensitivity, especially in some subgroups), I see more diagnostic delays, as clinicians are more hesitant to make a clinical decision. Hence, my comment still stands. 

I agree with you that clinical decision making may mask the impact of the implementation of more sensitive diagnostics, and that timely clinical diagnosis of TB might reduce the possibility of diagnosing eg RR-TB. Indeed, a more exhaustive evaluation, which in decentralized setting often implies referral of the patient or transport of samples to a central lab, will result in more accurate diagnosis, including resistant TB. However, in patients with an initial negative TB test, and with underlying TB disease, such diagnostic delays also may result in more advanced TB disease, pre-treatment attrition, and continued transmission. 

I therefore recommend to show a more balanced view on pro's and con's of clinical decision making for the diagnosis of TB. 

Reviewer #5: This paper investigates the factors influencing clinicians' decisions to initiate tuberculosis (TB) treatment after a negative bacteriological test result, a situation that occurs in over one-third of global TB diagnoses. Using a systematic review and individual patient data meta-analysis, the study analyzed trials and cohort studies conducted from 2010 to 2022. It found that several factors increased the likelihood of treatment initiation, including male sex, a history of prior TB, symptoms like cough and night sweats, and HIV infection without ART. However, clinicians were less likely to start treatment following a negative result from more sensitive, PCR-based diagnostics like Xpert MTB/RIF, compared to traditional smear microscopy.

I believe the authors have effectively addressed the reviewers' comments and concerns. However, I do have a couple of additional questions.

1. Since Bayesian methods were used, I didn't notice any mention of the burn-in period for the posteriors. Could the authors clarify the burn-in they used?

2. Additionally, have the authors checked the trace plots of the coefficients to ensure that the estimates have fully converged?

[LINK]

REQUESTS FROM THE EDITORS:

- Thank you for providing a completed PRISMA checklist. Could you please revise the right most column, replacing the page numbers with paragraph/section and line numbers? This is because page numbers are subject to change during the publication process. Apologies for not highlighting this requirement beforehand.

- Thank you for providing us with an Author Summary. This section, as it stands, needs a small revision. In the final bullet point of ‘What Do These Findings Mean?’, please describe the main limitations of the study in non-technical language.

- In the last sentence of the Abstract Methods and Findings section, please describe the main limitation(s) of the study's methodology.

- In your Abstract, please combine the Methods and Findings sections into one section: i.e. ‘Methods and findings’.

- Please add the following statement, or similar, to the Methods section: ‘This study is reported as per the Preferred Reporting Items for Systematic Reviews and Meta-Analyses (PRISMA) guideline (S1 Checklist).’

---

## [Decision Letter · Decision Letter 3]

20 Nov 2024

Dear Dr Kim, 

On behalf of my colleagues and the Academic Editor, I am pleased to inform you that we have agreed to publish your manuscript "Factors associated with tuberculosis treatment initiation among bacteriologically negative individuals evaluated for tuberculosis: an individual patient data meta-analysis" (PMEDICINE-D-24-01238R3) in PLOS Medicine.

PRESS

Sincerely, 

Syba

Syba Sunny, MBBS, MRes, FRCPath 

Associate Editor 

PLOS Medicine